# The Enigma of Neuromelanin: How Animal Models Can Help Researchers to Understand the Link Between Pigments and Neurodegeneration

**DOI:** 10.3390/biomedicines13123048

**Published:** 2025-12-11

**Authors:** Svenja Esser, Maximilian Hausherr, Katrin Marcus-Alic

**Affiliations:** 1Medizinisches Proteom-Center, Medical Faculty, Ruhr University Bochum, 44801 Bochum, Germany; 2Medical Proteome Analysis, Center for Proteindiagnostics (PRODI), Ruhr University Bochum, 44801 Bochum, Germany

**Keywords:** neuromelanin, neuromelanin granules, substantia nigra, Parkinson’s disease

## Abstract

Neuromelanin granules (NMGs) are the characteristic hallmark of the human *substantia nigra pars compacta* (SNpc). Although especially NMG-containing neurons are affected by neurodegeneration in Parkinson’s disease (PD), the complex biology of NMGs is not fully understood. This is partly due to the limitation that classical animal model systems do not produce NMGs. In this review, we summarize the plethora of animal models used to investigate NMG formation. While these models have already contributed to our knowledge about pathomechanisms in PD, especially in strengthening the interplay between lysosomal dysfunction, accumulation of NMGs and degeneration of dopaminergic neurons, open questions remain. However, new animal models combining NMG formation and PD-like pathology promise to offer immense value for studies on novel therapeutic approaches for PD.

## 1. The Complex Biology of Neuromelanin

The enigmatic nature of the black-brownish pigment neuromelanin (NM) has been a research topic since its first description in the 19th century [1]. Since then, the questions about the origin of the pigment itself as well as its function remained burning in the scientific community. Especially during the 20th and early 21st century, new findings added information but could not definitely answer these questions. NM accumulates over life mainly in the *substantia nigra* (SN) and *locus coeruleus* (LC) but can also be found in other brain regions to a lesser extent. Additionally, there are areas in the labyrinth, inner ear and meninges which are pigmented [2,3,4].

The research efforts covering NM were intensified after a key finding by Hirsch et al., showing that especially NM-rich dopaminergic neurons in the *substantia nigra pars compacta* are lost during Parkinson’ s disease (PD) [5]. Recently, the decrease in NM-content has been extensively studied as an imaging biomarker in MRI of the so-called nigrosome, supporting diagnosis of PD [6,7,8]. In *post-mortem* tissue, the neuronal loss is even visible to the naked eye due to a depigmentation of the usually black-brownish *substantia nigra* and its observation raised the question, how NM is either contributing to or preventing neurodegenerative processes in PD.

Such questions would usually be addressed in different model systems, such as cell culture lines or animal models. Unfortunately, NM is neither present in animal models like mice or rats, nor in frequently used neuronal cell culture models like SH-SY5Y cells. Therefore, most of what is currently known about NM originates from studies on human *post-mortem* tissue, which have several limitations. Such tissue samples only display a certain time point and this do not allow longitudinal observations over the course of a disease. Additionally, they are heavily requested amongst researchers, limited in availability, and can only be used when stringent ethical criteria are met.

Still, studies based on human *post-mortem* tissue have successfully clarified biochemical properties of the pigment NM itself as well as numerous aspects regarding the neuromelanin granules (NMGs), organelle-like structures in which NM is stored intracellularly. NM is characterized by its black-brownish color, which originates from NM being a mixture of eu- and pheomelanin [9]. In addition to its melanin content, NM consists of a substantial proportion of proteins that are described to be incorporated in its 3D-structure [10]. Further, NM is reported to show a high affinity to ferrous and ferric iron, as well as other metal ions and environmental toxins like MPTP [11,12]. These characteristics are often used as an explanation for the dualistic nature of NM, showing both protective and potentially toxic properties, probably after its binding capacities are saturated [13].

The NM containing NMGs are characterized by a surrounding double membrane, which shields their content from the cytosol [14]. Besides NM and proteins, NMGs contain a high degree of lipids, especially dolichol [15,16], but also glycerophospholipids, sphingolipids, and cholesterols [17,18]. Shortly after their discovery, it was hypothesized that NMGs may be (auto)lysosome-related structures and that their cargo was therefore intended for degeneration [18]. Given that the number of NMGs increases in with age and that NM pigmentation becomes more intense [19], this suggests that NMG accumulation reflects a progressive decline in the efficiency of the endo-lysosomal system to degrade the NMG cargo. Such processes are known from lysosomal-storage disorders [20] and would further explain why genetic risk factors for PD are often linked to lysosomes (e.g., *LRRK2*, *GBA*) [21]. Although substantial experimental evidence supports the connection between NMGs and lysosomes, definite experimental proof is still lacking, as recently emphasized in a review article [22]. Such an experiment would require monitoring the intracellular generation of NM and the subsequent formation of NMGs. However, these investigations cannot be conducted using *post-mortem* brain tissue and therefore necessitate appropriate cell culture or animal models.

Since a recent review has comprehensively summarized the neuropathological features of NM accumulation in dopaminergic and noradrenergic neurons [22], we will focus here on the remaining open research questions and how these can be addressed using animal models.

## 2. Which Research Questions About Neuromelanin Need to Be Addressed?

Currently, three major research areas require further experimental investigation: while the process of NMG formation is the most apparent topic in need of experimental validation, the mechanisms underlying NM synthesis and the role of NMGs in neurodegenerative processes also remain incompletely understood.

For the synthesis of NM, either an enzyme-free or an enzyme-mediated mechanism are discussed. The enzyme-free mechanism is based on the auto-oxidative nature of dopamine, which must be prevented in dopaminergic neurons by maintaining dopamine at a low pH within synaptic vesicles [23]. It is proposed that, when this storage process is disturbed, NM forms reactive intermediates (e.g., dopamine quinone), which undergo further auto-oxidation and can also interact with cytosolic macromolecules [24]. Finally, such auto-oxidative reactions result in the formation of melanic compounds that are, however, reported to be structurally less complex than NM [25]. Moreover, if neuromelanin were formed solely through non-enzymatic auto-oxidation of dopamine, similar pigment formation should occur in the dopaminergic neurons of mice and rats, yet as noted earlier, no such neuromelanin accumulation is observed in these species. Contrary to the auto-oxidation hypothesis, which is largely supported by cell culture studies showing L-DOPA-induced melanin accumulation in PC12 cells [26], alternative mechanisms involving the enzymes tyrosinase (TYR) or tyrosine hydroxylase (TH) have been proposed. It should be noted, however, that a TH-based mechanism could also involve an auto-oxidative component [27], since TH is essential for dopamine synthesis and is therefore highly expressed in all neurons containing NMGs. The most often discussed hypothesis, however, involves NM synthesis by TYR. This enzyme is also involved in melanin synthesis in human skin, where L-DOPA serves as its initial substrate [28], making it tempting to also assume a similar mechanism for NM production. Although several studies have reported the presence of TYR mRNA in the human *substantia nigra*—the brain region with the highest NM content—corresponding evidence at the protein level. remains lacking [18,29,30,31,32,33,34,35]. To date, convincing reports of TYR protein expression in the human brain are absent, casting doubt on the role of TYR in NM synthesis. Nevertheless, overexpression of the human TYR gene has been shown to induce NM-like pigment formation in both cell culture and animal models, a result that cannot be replicated by, e.g., TH overexpression

Similarly to the questions regarding the function of NMGs, those concerning their role in neurodegenerative processes would immensely benefit from time-resolved experimental approaches. While extracellular NM, as observed in PD-affected brains, is known to activate surrounding microglia, it remains unclear whether NM is actively released from neurons during degeneration or merely persists after the degeneration of the originally containing neuron. A recently postulated theory suggests the existence of an intracellular threshold for NMG accumulation, which, once exceeded, triggers neuronal degeneration and subsequent release of NMGs into the extracellular space [13]. The mechanism may also be initiated by other pathogenic processes associated with PD, such as elevated oxidative stress or the presence of aggregating proteins in the form of Lewy bodies. Although such theories provide valuable conceptual frameworks and stimulate scientific discussion, they are difficult—if not impossible—to verify or falsify solely on the basis of human *post-mortem* tissue. This urgent need for experimental models to study NM generation and NMG formation has therefore become increasingly recognized, leading to the development of diverse cellular (briefly discussed in this review) and animal models. These models (summarized in Figure 1) are heterogeneous in their design and generation methods, offering distinct research opportunities for the scientific community. A detailed summary of the studies considered for this review is provided in Table 1.

## 3. Which Models Exist?

### 3.1. Exogenous NM Application—Synthetic NM

The idea that NM forms a byproduct of dopamine (DA) oxidation is supported by experiments showing that NM-like pigments can be synthesized in vitro by oxidation of DA under basic conditions, with or without addition of L-cysteine [25,54,58,59]. In some cases, synthetic NM has been enriched with iron, reflecting the known ability human NM to bind redox-active Fe ions

Numerous studies utilizing synthetic pigments—often referred to as DA-melanin—in both cell culture and rodent models. In immortalized cell lines such as PC12 or SH-SY5Y, exposure to DA-melanin has been associated with apoptosis, mitochondrial dysfunction and increased ROS [60,61,62,63]. In rodent models, synthetic NM is commonly introduced by stereotaxic injection into the *substantia nigra* (SN) or other brain regions (*striatum*, *locus coeruleus*). These studies consistently report that exogenous NM triggers neuroinflammatory responses, dopaminergic neurodegeneration and motor impairments, thereby recapitulating some pathological hallmarks observed in PD [49,50,54].

Recently, more complex NM-structures have been synthesized to more closely resemble their human counterparts. Liu et al. developed poly-DA (pDA) nanoparticles designed to mimic the two major melanin species present in human NMGs—eumelanin or pheomelanin. These nanoparticles were loaded with Fe and applied to SH-SY5Y cells as well as rodent brains to investigate the neurotoxic and neuroprotective properties of each melanin type [49]. Given recent findings that NMGs in PD patients are enriched in pheomelanin [64], and that eu- and pheomelanin differ in their redox behavior and iron-binding capacities [65], studying these species individually provides valuable insight into their distinct roles in NM-related pathology.

Additionally, aminochrome (AC), the most stable intermediate in the biochemical pathway leading from DA oxidation to NM formation, has been identified as a potential key mediator of neuronal toxicity. AC formation occurs physiologically in dopaminergic neurons; however, under certain conditions, AC can be enzymatically reduced to highly neurotoxic semiquinone species and can form adducts with alpha-synuclein [66,67]. These properties make AC an attractive research target both as a potential contributor to neurodegeneration and as a molecular trigger to simulate PD-like pathology in experimental models. Similarly to NM, AC can be synthesized in vitro and subsequently applied to cell culture or rodent models [52,53,68,69].The addition of AC induces dysfunction within the dopaminergic system, including morphological alterations in TH-positive neurons, increased γ-aminobutyric acid (GABA) release, and impaired neuronal vesicular transport—features commonly associated with PD pathology [53]. AC is considered a more physiologically relevant alternative to conventional neurotoxins such as rotenone or MPP+, given its slower mode of action and its origin as a physiological byproduct of DA metabolism.

### 3.2. Exogenous Application—Bacterial Melanin

In addition to fully synthetic NM, several bacterial strains are capable of producing melanin-like pigments. Petrosyan et al. investigated melanin derived from a *Bacillus thuringiensis* strain for its neuroprotective potential following unilateral destruction of the SN in rats [48]. Earlier studies had already demonstrated beneficial effects of bacterial melanin on neuronal regeneration after surgical injury. Intramuscular administration of bacterial melanin was shown to alleviate surgery-induced motor deficits, suggesting a positive influence of melanin or NM on neuronal survival and development [70]. Beyond its neurobiological effects, bacterial melanin exhibits several practical advantages: it is easy to handle, cost-efficient to produce, a more water-soluble than melanin synthesized from DA or isolated from human tissue, and capable of crossing the blood-brain-barrier, properties that make it an attractive candidate for neuroprotective applications [48]. More recently, a genetically engineered *Escherichia coli* (*E. coli*) strain has been introduced as a novel source for NM-like pigments. The neuroprotective effects of this *E. coli*-derived melanin were evaluated both on MPP+-treated SH-SY5Y cells and mice. In these models, treatment reduced astrocyte activation and restored dopaminergic neuronal signaling. In vivo, administration of *E. coli*-derived melanin also mitigated MPP+-induced motor deficits, further supporting the potential of bacterially synthesized melanin for therapeutic research [47].

Although bacterial melanin represents a cost-efficient and experimentally accessible source of NM—like material, it has been used far less frequently than in vitro generated melanin species. The endogenous biosynthesis of melanin within living organisms may enable more complex pigments, incorporating lipids and proteins in a manner similar to human NMGs—features that cannot be replicated by DA oxidation under laboratory conditions. However, despite the promising findings regarding its potential neuroprotective properties, an in-depth biochemical characterization of bacterial NM is still lacking, particularly with respect to protein and lipid composition and its structural resemblance to human NM.

### 3.3. Exogenous Application—Human NM

The use of human NMGs in animal models of PD is less common, primarily due to ethical constraints and the limited availability of non-fixed *post-mortem* tissue. Nevertheless, the first in vivo studies investigating NM-related neurodegeneration in animal models were conducted using human NMGs. In 2003, Double et al. introduced iron-loaded human NMGs into the SN of rats via stereotaxic injection to assess the neuroprotective potential of lisuride, a dopamine receptor antagonist [57]. Similarly, Zecca et al. injected human NMGs into the rat SN and observed characteristic degeneration of TH-positive neurons within the injected site along with pronounced microglia activation [56], which was also observed in the third study utilizing human NMGs in rats [55].

Exogenous application of synthetic or human NMGs enables precise temporal and spatial control over NMG exposure, allowing defined amounts to be administered at specific time points. In the human brain, however, this approach does not replicate the gradual, age-dependent accumulation of NM and that occurs in the human brain. In humans, NMGs build up progressively over decades, accompanied by a continuous increase in NM content. In contrast, a single injection into rodent brain induces an acute exposure rather than a slow deposition process, thereby limiting the model’s suitability for studying the chronic and progressive nature of PD.

### 3.4. Endogenous NMG Formation—Animal Models with AAV-Mediated hTYR Expression

To overcome the limitation of the absent endogenous NMG formation in both cell culture and rodent models, overexpression of TYR, the key enzyme in peripheral melanin biosynthesis, has proven to be an effective strategy. Hasegawa et al. were first to induce NMG formation in a genetically modified clone of the widely used SH-SY5Y neuroblastoma cell line by transfecting the cells with a plasmid carrying the human tyrosinase (hTYR) gene under control of a tetracycline-controlled promotor system [71]. The inducible setup allows for tightly controlled experimental conditions. Upon tetracycline-induced hTYR expression, the cells progressively accumulate NMG-like pigmented structures. predominantly localized within lysosomal organelles, which also contain the TYR enzyme [71]. Furthermore, proteomic comparison between human NMGs and those generated in hTYR-expressing SH-SY5Y cells revealed a high degree of similarity in protein composition. (*unpublished data*, 2025), supporting the validity of this system as a promising in vitro model of NMGs. More recently, hTYR expression has also been achieved in more complex in vivo model systems, enabling the study of NM formation and accumulation under physiological conditions.

In 2019, Carballo-Carvajal et al. introduced the first genetically modified rodent model capable of endogeneously producing NMG-like structures through overexpression of hTYR. In this model, rats were transfected via stereotaxic injection of an adeno-associated viral (AAV) vector carrying the hTYR gene under the control of a cytomegalovirus (CMV) promoter into the right SN. Following injection, formation of a dark pigment was observed near the injected site, which progressively accumulated in TH-positive neurons to levels comparable to those seen in the SNpc of elderly humans [46]. The gradual accumulation of hTYR-induced NMGs was accompanied by progressive dopaminergic neurodegeneration, neuroinflammation, formation of inclusion bodies, and a marked decrease in TH immunoreactivity [46]. This transgenic rodent model therefore was the first to recapitulate an age-dependent, endogenous accumulation of NMGs in vivo. In subsequent years, several studies have employed the hTYR-overexpression model in rodents or non-human primates to investigate both neurotoxic and neuroprotective mechanisms associated with NMG accumulation [36,37,38,39,40,41,42,43,44,45].

More recently, this model has been further refined to achieve a more selective expression and physiologically relevant expression patterns. By placing the hTYR gene under the control of the TH promoter, expression was restricted to catecholaminergic neurons throughout the brain [41]. In addition, the use of modified AAV capsids capable of efficiently crossing the blood-brain-barrier has eliminated the need for stereotaxic surgery, thereby improving reproducibility and reducing animal stress. This latest generation of hTYR-based mouse models faithfully recapitulates the age-dependent pigmentation of the SNpc, LC and VTA (ventral tegmental area) along with features of PD pathology [36].

Although hTYR expression currently represents the only successful strategy to induce NMG formation in rodents, non-human primates, and immortalized cell lines and effectively reproduces several key hallmarks of PD, the presence of TYR in the human brain is not proven, as previously mentioned. Consequently, hTYR-based models are not suitable for elucidating endogeneous biosynthetic pathway of NM. Further, as observed for bacterial melanin, a comprehensive biochemical characterization of NMGs produced via hTYR overexpression in rodent brains, particularly regarding their protein and lipid composition and their similarity to human NMGs, is still lacking.

To date formation of NMGs without genetic manipulation has only been demonstrated in midbrain-like organoids derived from human embryonic or induced pluripotent stem cells (iPSCs). Upon prolonged culture, either natural aging over two months or artificial aging induced by progerin, these organoids began to form pigmented structures that increased in number over time and stained positively for melanin using the Fontana-Masson staining method. The authors of these studies postulated that such systems provide a valuable platform for investigating NMG biosynthesis and accumulation [72,73]. Further, in dopaminergic neurons differentiated from iPSCs from PD patients with a mutation in the *PARK7* gene as well as in dopaminergic neurons differentiated from iPSCs from healthy controls with a CRISPR-CAS9 mediated knock-out of *PARK7*, NM production was observed resulting from elevated dopamine oxidation after an extended time period in culture [74].

Nevertheless, the limited number of publications employing human midbrain-like organoids likely reflects the technical complexity and high cost associated with this approach. Despite these limitations, organoid systems derived from human stem cells offer unique advantages, including highly physiological cellular environment, compatibility with personalized medicine approaches using patient-derived iPSCs, and a reduction in the use of experimental animals. However, organoids still lack the structural and functional complexity of the intact brain, particularly the intricate interplay between distinct brain regions. Since NM and NMGs are not confined to dopaminergic neurons of the SNpc, animal models capable of endogenously generating NMGs continue to provide superior systems for studying the broader biology of NM and NMGs.

### 3.5. Combination with Other Pathological Features of PD

PD is a disease multiplies characterized by multiple pathological hallmarks, including increased oxidative stress, mitochondrial dysfunction, accumulation of ɑ-synuclein (ɑSyn)-containing aggregates (Lewy bodies, LB), and impaired dopamine signaling. The exact molecular mechanisms underlying the selective vulnerability of pigmented dopaminergic neurons in PD remain incompletely understood, particularly in sporadic cases that are not linked to mutations in specific genes. To advance our understanding of the biological processes leading to dopaminergic neurodegeneration and the potential contribution of NMGs to these processes, it is essential to establish in vivo models that recapitulate multiple key features of PD pathology.

The deposition of Lewy bodies is one of the key hallmarks in PD, and NMGs have been shown to incorporate ɑSyn early in disease progression [75]. Although animal and cellular models with genetic mutations in the *SNCA* gene locus, leading to overexpression or aggregation of the protein ɑSyn, are widely used [76,77], only a limited number of studies have attempted to combine ɑSyn pathology and NMG accumulation. Recently, Garcia-Gomara et al. developed a mouse model integrating both of those key hallmarks by stereotaxic delivery of an AVV vector encoding hTYR into mice expressing human ɑSyn. This combined model provides an advanced platform to investigate the mechanistic interplay between NMG accumulation and LB pathology in PD [37].

Based on the hypothesis that NM originates as an oxidative byproduct of cytosolic, nonvesicularly stored DA, overexpression of the vesicular monoamine transporter 2 (VMAT2) has been proposed as a potential therapeutic option to limit NM accumulation [26,78]. VMAT2 mediates the sequestration of DA into synaptic vesicles, thereby reducing cytosolic DA levels and its susceptibility to auto-oxidation. In two independent studies employing either a SN-derived cell line, PC12 cells, or rat SN slice cultures, overexpression of VMAT2 resulted in reduced NMG formation and lower cytotoxicity [26,69]. Recently, this protective effect was evaluated in vivo using previously described hTYR-overexpression mouse model. In this system, co-expression of VMAT2 mitigated pigment accumulation and neuronal stress, supporting the notion that efficient vesicular DA packaging counteracts NM buildup [45]. Notably, *post-mortem* studies have shown that VMAT2 expression is reduced in the SNpc of PD patients, further reinforcing the link between impaired vesicular storage of DA, elevated cytosolic DA, and enhanced NM formation.

The introduction of rodent models with endogenous NMG production has led to a noticeable increase in experimental research on NMGs and its potential role in PD pathogenesis, with at least eight publications emerging since 2019 and several preprints currently available on platforms such as *bioRxiv*. Animal models that combine multiple PD-related features—such as ɑSyn aggregation, oxidative stress, and impaired DA handling—offer more physiological framework to investigate the complex interactions between NMGs and molecular features. Understanding these interactions is essential for elucidating proteins or their effect on important biological processes and are still highly demanded to further advance the contribution of NMGs to dopaminergic neurodegeneration and for developing potential therapeutic strategies. The recent methodological advancements by Chocarro et al., introducing hTYR-encoding viral vectors that can cross the blood–brain barrier without stereotaxic surgery, represents a significant step forward [36]. This approach enables more efficient, standardized, and combinatorial modeling—allowing integration of multiple pathological and therapeutic components within a single animal system—and is likely to become a valuable platform for future studies on NM biology and PD treatment development.

## 4. What Can We Learn from the Studies Conducted So Far?

The studies discussed above provide valuable insights into the potential contribution of neuromelanin granules (NMGs) to the pathogenesis of Parkinson’s disease (PD). The key findings and concepts derived from these investigations are summarized in the following paragraphs and illustrated in Figure 2.

### 4.1. Insights into Underlying Biological Processes of Neurodegeneration Drawn from In Vivo Models

The involvement of NMGs in the neurodegeneration of dopaminergic neurons in PD has been debated for decades [79]. However, the lack of suitable models capable of reproducing the age-dependent accumulation of NMGs limited experimental evaluation of this hypothesis. With the development of animal models enabling endogenous NMG formation via hTYR overexpression, NMG accumulation could be directly linked to progressive dopaminergic neurodegeneration and the onset of typical parkinsonian symptoms such as motor deficits and behavioral alterations, in rodents and non-human primates [43,46]. Importantly, progressive NMG accumulation in the SN led to the deposition of Lewy Body-like structures, mainly in pigmented neurons. This observation mirrors one of the defining pathological hallmarks of PD and is in-line with findings from human post-mortem studies, in which ɑSyn was shown to be co-localized with NMGs [75] and to be enriched within them relative to surrounding tissue [33]. Such aggregation may result from a gradual breakdown of proteostasis and impaired autophagic clearance following progressive intracellular crowding with NMGs. Indeed, dysfunction of protein homeostasis and autophagy is widely recognized as a central contributor to PD pathogenesis and several genes implicated in familial PD encode proteins associated with the autophagy-lysosomal machinery [80].

From these and other findings, a pathological threshold hypothesis has been proposed [13,46]. According to this concept, dopaminergic neurons begin to exhibit key pathological alterations—such as neuroinflammation, mitochondrial dysfunction, and impaired proteostasis—only once the intracellular concentration of neuromelanin granules (NMGs) exceeds a critical threshold. Supporting this notion, post-mortem analyses in patients with PD revealed elevated neuromelanin content in dopaminergic neurons prior to the onset of overt neurodegeneration [75]. It has therefore been suggested that the cellular attempt to sequester and degrade accumulating NM within autophagic vesicles may progressively overload of the cells’ proteostasis machinery. This overstrain could trigger the autophagy-lysosomal system, promote a-syn aggregation, and ultimately lead to neuronal death [46]. Although an age-dependent build-up of NMGs in SNpc neurons is also observed in humans, only ~1% of people above the age of 65 develop PD. The discrepancy has led to the hypothesis that PD patients may reach the pathological threshold earlier in life due to accelerated NMG formation, potentially driven by dysregulation of other enzymes such as L-dopachrome tautomerase or VMAT2 [46].

Magnetic resonance imaging (MRI) studies of NM in both hTYR expressing and patients with PD or isolated rapid eye movement sleep behavior disorder (iRBD)—a prodromal stage of the disease—revealed a comparable pattern of longitudinal changes in NM concentration within the SNpc of hTYR expressing rats and humans prior to the onset of neurodegeneration. These findings provide additional support for the hypothesis that exceeding a critical threshold of NMG plays a pivotal role in triggering dopaminergic cell loss in PD [42]. Moreover, this study generated important mechanistic insights into the biophysical origin of NM-MRI signal changes, which is still debated. The strong correlation between intracellular NM accumulation in hTYR-expressing rats and the T1 reduction effect observed on MRI suggests that the contrast alterations detected by NM-MRI are indeed driven by T1-shortening properties of NM itself, rather than by secondary factors such as iron content or tissue microstructure [42].

Another key contributor to the pathology of PD is neuroinflammation [81]. Strikingly, activation of microglia has been consistently reported across most cellular and animal studies in the context of NM, NMGs or AC. When human NMGs were stereotaxically introduced into the SN of rats, pronounced microgliosis was observed as early as one week post injection [56]. Several studies have since detected extracellular NM and NMGs inside microglia, indicating that these immune cells actively phagocytose NM and NMGs derived from degenerating neurons—an observation corroborated by histological analysis of human post mortem tissue [55,82,83]. More recently, Tejchman-Skrzyszewska et al. used synthetic NM injected into mouse SN to investigate the immune response to extracellular NM, demonstrating not only microglia activation but also astrogliosis. Reactive astrocytes secrete pro-inflammatory mediators that act on microglia, thereby amplifying neuroinflammatory signaling and microgliosis [50]. These findings suggest a self-perpetuating cycle of glial activation that may exacerbate dopaminergic neurodegeneration. Introduction of exogenous NM, whether of human, bacterial or synthetic origin, primarily models extracellular accumulation of NM, rather than the intracellular NMG crowding characteristic of dopaminergic neurons in PD. The mechanisms by which NM is released from neurons into the extracellular space remain unresolved. To elucidate this process, model systems with endogenous NM formation are essential, as they better recapitulate the physiological dynamics of NM turnover and release during neuronal stress or degeneration.

Studies using hTYR-overexpressing rodent models revealed that neuroinflammatory responses mainly occur after the onset of dopaminergic neurodegeneration, indicating that extracellular NM released from dying neurons serves as a principal trigger of microglial activation. In an advanced model combining human αSyn overexpression with, hTYR-mediated NMG formation, Garcia-Gomara et al. identified the co-chaperone FKBP51 as a key regulator of both neuroinflammation and αSyn aggregation through in-depth transcriptomics analyses [37]. FKBP51 is known to modulate stress responses, glucocorticoid receptor signaling, and activation of inflammatory microglia, linking it to inflammatory and stress-related pathways implicated in PD. Pharmacological inhibition of FKBP51 using SAFit2 significantly reduced microgliosis, αSyn-positive inclusions, and NM accumulation in dopaminergic neurons. Interestingly, *FKBP5* gene expression is also upregulated with aging in the human brain, providing a potential mechanistic link between age-dependent vulnerability, NMG accumulation, and inflammatory stress responses [37,84,85]. These results highlight FKBP51 and its molecular interactors as promising pharmaceutical targets for mitigating both neuroinflammation and protein aggregation in PD.

Beyond their accumulation and their role in neurodegeneration, many questions remain around NM biosynthesis and its function. As mentioned before, NM consists of two chemically distinct melanin species—eumelanin and pheomelanin—which differ in their antioxidant potential, photostability, and iron-binding capacity. Recent analyses of NMGs isolated from post-mortem SN tissue of PD patients revealed a selective enrichment of pheomelanin compared to healthy subjects and subjects with Alzheimer’s disease (AD) [64]. These findings led to the hypothesis that pheomelanin and eumelanin differentially influence dopaminergic vulnerability. Supporting this notion, synthetic pheomelanin significantly reduced cell viability in differentiated SH-SY5Y cells and mouse primary neurons, whereas eumelanin exerted no comparable toxic effect [49]. Liu et al. postulated that the stronger redox-activity and iron-binding capacity of pheomelanin make it a double-edged contributor to neuronal homeostasis—potentially neuroprotective under physiological conditions, but neurotoxic in the context of iron overload or oxidative stress [49]. However, NM synthesized in absence of cysteine, yielding eumelanin, utilized in other studies, did show neurotoxic effects, and melanin produced in the leptomeninges, also thought to be eumelanin, is discussed to be a source for aseptic dermatitis [2]. Given those implications, the distinct roles of eu- and pheomelanins in neuronal vulnerability still need in-depth validation. Despite these advances, the biosynthetic pathway of NM in the human brain is not fully understood, and no current animal model allows for the selective induction of pheomelanin synthesis in vivo. Elucidating the molecular determinants that govern NM composition will therefore be crucial to understand its dualistic role in neuronal resilience and degeneration.

### 4.2. Implications for Therapeutic Options and Future Directions Drawn from In Vivo Models

PD is the second most prevalent neurodegenerative disease, with rising prevalence over the past decades [86]. Consequently, in addition to elucidating the underlying mechanisms disease, one burning topic in medical research lies in the development of novel therapeutic strategies. Since the establishment of rodent models with endogenous NMG formation and the demonstrated correlation between progressive NMG accumulation and dopaminergic neurodegeneration, several studies have explored approaches aimed at preventing neurons from exceeding the proposed intracellular NMG threshold. Exceeding this threshold is thought to trigger proteostatic failure and subsequent neuronal death. Already in the initial 2019 publication describing hTYR-overexpressing rodents, it was postulated that enhancing autophagy activity could represent a promising therapeutic strategy to limit NMG accumulation below the pathological threshold, maintain neuronal proteostasis, and thereby attenuate PD progression [46]. This concept highlights the potential of targeting intracellular clearance pathways as a means to modulate NMG dynamics and counteract neurodegeneration in PD.

Another promising therapeutic avenue involves modulating VMAT2 expression in the brain. VMAT2 is responsible for sequestering DA into synaptic vesicles, thereby preventing its oxidation in the cytosol. The long-standing theory that inefficient vesicular packaging of DA accelerates NM formation through cytosolic DA oxidation has been substantiated by multiple experimental observations. Neuroprotective effects of VMAT2-overexpression in neuronal cells against AC-induced toxicity had already been demonstrated in neuronal cell culture models [69]. More recently, overexpression of VMAT2 in hTYR-overexpressing rats was shown to slow down NMG accumulation and delay the onset of motor symptoms, supporting the idea that therapeutic enhancement of VMAT2 activity may help limit NM buildup and mitigate dopaminergic neurodegeneration [45]. Thus, pharmacological or gene therapy-based modulation of VMAT2 represents an attractive direction for future PD treatment strategies.

In addition to genetic and molecular interventions, transcranial focused ultrasound, a technique used to prevent hyperpigmentation of the skin as well as amyloid plaque formation in AD models,—has been evaluated as non-invasive physical approach to reduce pathological pigmentation. Previously used to limit hyperpigmentation in the skin [87], transcranial focused ultrasound was recently applied to hTYR-overexpressing rodents. Remarkably, treatment during early stages of NMG accumulation effectively reduced NMG load, accompanied by decreased Lewy body deposition and attenuated dopaminergic neurodegeneration. However, these protective effects were only observed when intervention occurred during early or prodromal stages, underscoring the importance of timing in potential future therapeutic applications [44].

Garcia-Gomara further explored anti-inflammatory strategies by utilizing hTYR-expressing mice to assess the protective effects of the glucocorticoid dexamethasone, a potent inhibitor of microglial activation and proliferation. Treatment with dexamethasone resulted in a reduction in neuroinflammation during early stages of NM-induced, PD-like neurodegeneration and was accompanied by and amelioration of motor deficits. Notably, the therapeutic effect of dexamethasone was most pronounced at the peak of dopaminergic neuron loss, where dexamethasone was able to attenuate, but not fully prevent progression of neurodegeneration [38]. These findings highlight that targeting neuroinflammation may slow disease progression, but is insufficient as a standalone strategy once extensive neuronal loss has occurred.

PD exhibits a higher prevalence in males, and clinical symptoms differ between sexes [88,89], emphasizing the importance of understanding sex-specific factors in disease onset and progression to develop personalized therapeutic strategies. Recently, D’Addario et al. investigated the sex-dependent effects of NMG accumulation over time. While in males, NM build-up, mitochondrial dysfunction, and the development of anxiety-like symptoms were observed, motor deficits occurred in both sexes. Functionality of dopaminergic neurons was altered in opposite directions, showing hyperactivity in females, and hypoactivity in males [39]. Although some molecular differences have been identified between male and females, the mechanistic basis for these discrepancies and the higher prevalence of PD in men remain largely unknown Given the strong correlation between NMG accumulation and degeneration of dopaminergic neurons, this study by D’Addario et al. provide valuable insights into sex differences and may help to elucidate the biological factors underlying PD susceptibility.

Animal models of NMGs have already provided important insights into the pathomechanisms of PD. In particular, the establishment of rodent models with endogenous NMG formation via hTYR expression has enabled researchers to link age-dependent NMG accumulation to degeneration of dopaminergic neurons and other classical hallmarks of PD. From those novel insights, several promising therapeutic targets have emerged, primarily focused on reducing NMG accumulation, formation, and associated neuroinflammation. However, to the development of effective treatments ultimately depends on a comprehensive understanding of the biological basis of both PD pathology and NMG biosynthesis.

## 5. Which Questions Remain to Be Addressed and How Can They Be Tackled?

Although recent advances—particularly the introduction of animal models with endogenous NMG production—have significantly deepened our understanding of NM and NMG biology, many fundamental questions remain unanswered. All currently available animal models rely either on overexpression of the enzyme TYR or on exogenous application of NM, yet the mechanism of NM biosynthesis in the human brain itself is still unknown. Consequently, these models cannot directly address this key aspect. To date, the only systems that allow for spontaneous NMG formation are human midbrain-like organoids derived from stem cells, which form NMGs naturally when cultured over a long time period. However, the cultivation of such organoids is technically demanding and can only be performed in specialized laboratories. Despite these limitations, they currently represent the most promising model for uncovering the mechanism underlying NM formation in the human brain.

Regarding the mechanism underlying NMG formation, hTYR-overexpressing animal models represent a promising experimental tool. Time-course studies tracking NMG formation it different stages could help to finally understand the connection of NMGs and the endolysosomal system, potentially revealing the subcellular processes involved in their biogenesis. Furthermore, live-cell imaging of acute brain slices, combined with fluorescent labeling of target proteins, may enable real-time observation of NMG formation and its cellular dynamics. Nevertheless, the biochemical characterization of NMGs produced in hTYR-overexpressing rodents remains incomplete. Their similarity to human NMGs should be assessed further through, e.g., proteomic and lipidomic analyses utilizing mass spectrometry-based techniques, comparing the molecular composition of both structures, or by other state-of-the-art analytical methods such as EPR or NMR spectroscopy. Such studies could be conducted after careful isolation of NMGs from rodent and human post-mortem tissue, either by density-gradient ultracentrifugation or laser microdissection-based workflows requiring smaller sample amounts.

Most studies utilizing animal models to date have focused primarily on the neurotoxic properties of NMGs, whereas their initially protective functions remain largely unexplored. Animal models with slow, progressive NMG accumulation offer the opportunity to analyze early molecular changes triggered by low intracellular NMG levels over time. Ideally, to minimize the use of laboratory animals, initial screenings could be conducted in cell models that mimic gradual NMG accumulation, allowing high temporal resolution at early stages of NMG formation. Results from such in vitro studies could then be validated in animal models. Furthermore, chronic co-exposure to additional cellular stressors may help to uncover potential protective functions of NMGs under physiological and pathophysiological conditions.

In summary, the establishment of advanced animal models for studying NM and NMGs has generated numerous valuable insights in a research field that, for decades, was constrained by a lack of suitable experimental systems. Although only six years have passed since the introduction of the first hTYR-overexpressing rodent model, several subsequent studies have highlighted the scientific significance and versatility of these models. Importantly, two promising therapeutic targets—VMAT2 and the lysosomal degradation system—have already been identified as potentially relevant for PD treatment. Moreover, recent MRI-based studies suggest that progressive NM and NMG accumulation may represent a critical risk factor for PD development. Understanding how this process can be modulated could therefore open avenues for early intervention strategies, potentially allowing prevention of PD onset even before dopaminergic neurons in the SNpc become affected.

Finally, it should be emphasized that reducing the number of animals used in scientific research must remain the central goal of the scientific community. Nevertheless, studies on NM and NMGs demonstrate that, due to the complexity of the brain, animal models are still indispensable for recapitulating the full physiological context. At the same time, the ongoing research into the mechanism of NM biosynthesis illustrates that organoid models already hold distinct advantages in specific aspects, highlighting their immense potential as complementary systems. As these technologies continue to evolve, they may ultimately bridge the gap between reduction in animal use and the need for physiologically relevant models in neuroscience research.

## Figures and Tables

**Figure 1 biomedicines-13-03048-f001:**
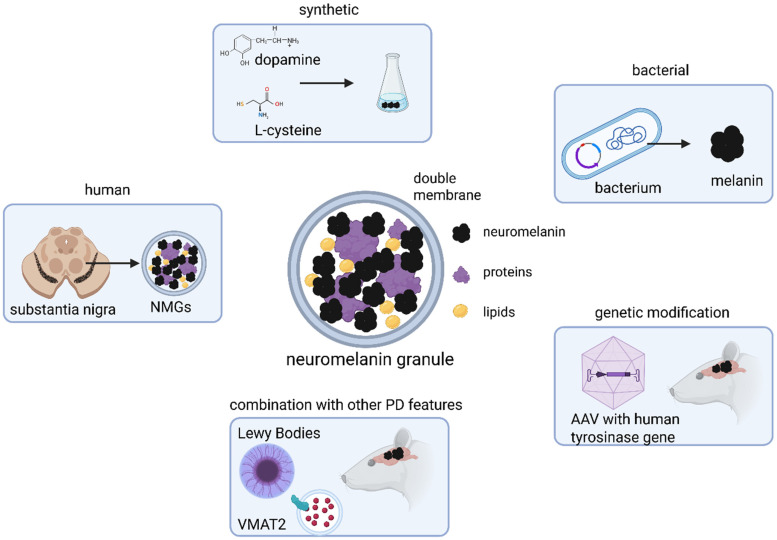
Sources for neuromelanin (NM) and neuromelanin granules (NMGs) used in animal models. Due to absence of endogenous NMG formation in common laboratory species, NM or NMGs are experimentally introduced into animal brains either by exogenous application or through genetic manipulation. Models employing stereotaxic injection typically use synthetic NM, produced in vitro via DA oxidation or produced by bacteria, or NMGs isolated from human brain tissue. In contrast, models of endogenous NM formation rely on adeno-associated viral (AAV) delivery of the human tyrosinase (TYR) gene into the rodent brain, thereby inducing neuromelanin synthesis in vivo. These genetically based have also been combined with other key features in Parkinson’s disease (PD) pathology, such as ɑ-synuclein (ɑSyn) aggregation or altered vesicular monoamine transporter 2 (VMAT2) expression, to generate more complex and disease-relevant models. Created in BioRender. Wulf, M. (2025) https://BioRender.com/g1cva15 (accessed on 4 November 2025).

**Figure 2 biomedicines-13-03048-f002:**
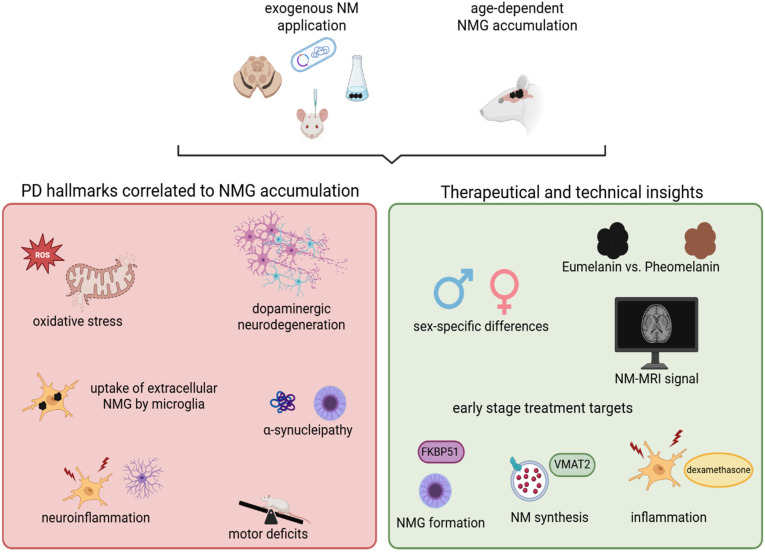
Animal models for NMGs have substantially advanced our understanding of the biological mechanisms underlying PD and have provided valuable implications for the development of novel therapeutic strategies. Key hallmarks of PD such as increased oxidative stress, degeneration of dopaminergic neurons, and neuroinflammation have been consistently reproduced across animal models of NM or NMGs. In rodents with endogenous NMG formation, the age-dependent accumulation of NMGs correlates with the onset of motor deficits and the emergence of Lewy body-like pathology. These models have also facilitated the identification of potential therapeutic targets, such as VMAT2 and FKBP51, which may be modulated during early disease stages. Furthermore, they offer unique opportunities to investigate sex-specific differences in PD pathology and to explore novel technical biomarkers relevant to disease diagnosis and progression. Created in BioRender. Wulf, M. (2025) https://BioRender.com/g1cva15 (accessed on 4 November 2025).

**Table 1 biomedicines-13-03048-t001:** Summary of considered publications on models for NM-/NMG-research.

Citation	Year	Animal	Type of Model	How was the Model Generated?	Key Results
Tyrosinase-Mediated Neuromelanin Generation					
Chocarro et al. [36]	2025	mouse	TYR	AAV-hTYR injection intravenously, BBB penetrable viral capsid	age-dependent NMG accumulation and neurodegeneration additional: pigmentation of the spleen, weak accumulation in sarcoplasmic cones of cardiomyocytes
Garcia-Gomara et al. [37]	2025	mouse	TYR	AAV-hTYR expression in mice expressing human asyn	age-dependent NMG accumulation leads to a decrease in TH neurons and neuroinflammation (Iba-1) 3 month old mice display inclusion bodies containing αSyn, p62, ubiquitin in pigmented neurons gene expression analysis: microglial activation, neuronal apoptosis, AD, neuronal development, increased number of microglia and serotonergic neurons in TYR animals, decrease in DA neurons and increased FKBP51 levels neurodegeneration of pigmented neurons was decreased via treatment with FKBP51 inhibitor SAFit2 one week after TYR induction
Garcia-Gomara et al. [38]	2024	mouse	TYR	AAV-hTYR overexpression	treatment with dexamethasone attenuates dopaminergic neuron loss, inhibits microglia activation, less behavioral deficits
D’Addario et al. [39]	2025	rat	TYR	AAV-hTYR injection bilateral	differences between male and female rats regarding rate of NM-build up, mitochondrial function, functionality of DA neurons, psychological symptoms
Iannitelli et al. [40]	2024	mouse	TYR	AAV-hTyr, Stereotaxic infusions, bilateral into LC	NM-like pigmentation of the LC leads to loss of fibers but not cell bodies, reduced NE and increased NE turnover (not DA), increased spontaneous pacemaker activity and evoked firing, upregulation of genes associated with ER stress response, prolonged accumulation induced neurodegeneration and neuroinflammation as well as astrocyte activation
Laguna et al. [41]	2024	mouse	TYR	hTyr expression under TH promotor: hTYR expression selectively in catecholaminergic neurons	Progressive pigmentation of catecholaminergic brain areas, highest levels in SNpc, VTA and LC, dopaminergic and noradrenergic neurodegeneration PD-like neuropathology, increasing amount of extracellular NM in aged mice, inflammation, impaired proteostasis, decrease in pigmented cells in DVC and autonomous dysfunction, decreased lifespan
Perot et al. [42]	2025	rat	TYR	AAV-hTYR expression in right side of SN	NM is released into extracellular space in both human and rats, similar pattern in NM-MRI signal changes in hTYR rats and humans from prodromal disease stages on initial build-up of NM followed by decrease due to neurodegeneration; correlation of MRI contrast to NM accumulation
Chocarro et al. [43]	2023	non-human primate	TYR	AAV-hTYR expression under CMV promotor, in left SN of four juvenile Macaca	pigmentation of the injected side, time-dependent loss of pigmentation, nigrostriatal neurodegeneration, more in female animals extracellular NM and NMGs in phagocytotic cells (microglia), formation of lewy bodies in pigmented cells and spreading towards prefrontal cortex, no motor symptoms observed
Compte et al. [44]	2023	rat	TYR	AAV-hTYR injection in right side of the brain, above SN	magnetic resonance guided transcranial focused ultrasound reduced intracellular and extracellular NM, decreased neuroinflammation and neurodegeneration, less motor symptoms when applied in early stages of NMG accumulation
Gonzalez-Sepulveda et. al. [45]	2023	rat	TYR	AAV-hTYR expression, AAV-VMAT2 overexpression	VMAT2 overexpression reduces NM formation and LB formation, prevents neurodegeneration
Carballo-Carbajal et al. [46]	2019	rat	TYR	AAV-hTYR expression, CMV promotor, unilateral in right side of SN	age-dependent NM accumulation in injected side, similar structure to human NM granules determined by EM, intracellular level similar to elderly humans progressive neurodegeneration of pigmented neurons, decreased DA, decreased TH, neuroinflammation, formation of inclusion bodies, also without αSyn
Models Based on Bacterial Melanin					
Kong et al. [47]	2024	mouse	bacterial	injection of E.Coli-derived melanin into MPTP mouse model	decreased degeneration of TH positive neurons, reduced astrocyte activation but not microglia activation by melanin motor impairments partially alleviated by melanin
Petrosyan et al. [48]	2014	rat	bacterial	unilateral destruction of SN, then injection of bacterial melanin	effects on motor recovery, regeneration of blood vessels in lesioned areas in melanin treated animals, no glial scar formed
Models Using Synthetic Melanin					
Liu et al. [49]	2024	mouse	synthetic	injection of pheomelanin and eumelanin-like nanoparticles with and without iron into mice SN	iron-loaded pheomelanin nanoparticles induced degeneration of TH neurons no degeneration without iron-loading pheomelanin without iron reduced impact of MPTP on motor symptoms
Tejchman-Skrzyszewska et al. [50]	2023	mouse	synthetic	injection of synthetic NM into SNpc	activation of microglia and astrocytes, upregulation of TH in longer treatment, loss of dopaminergic neurons, upregulation of pro-inflammatory cytokines
Mendes de Araujo et al. [51]	2023	rat	synthetic	injection of aminochrome and rutin or both into right side of the striatum	Rutin attenuates dopaminergic neurodegeneration and microglia activation induced by aminochrome, prevents astrogliosis
Mendes de Araujo et al. [52]	2023	rat	synthetic	injection of aminochrome into right side of the striatum	aminochrome induces dopaminergic neurodegeneration, microglia and astrocyte activation
Herrera et al. [53]	2016	rat	synthetic	injection of Aminochrome into left side of striatum	no significant neurodegeneration in SN and striatum, positive contralateral rotation decreased dopamine release and decreased number of synaptic vesicles isolated synaptic vesicles contained 1.6 fold more dopamine, higher number of damaged mitochondria and decreased ATP concentrations
Viceconte et al. [54]	2015	mouse	synthetic	intranigral neuromelanin injections	microglia activation in NM-injected animals determined by IHC
Models Based on Human Neuromelanin					
Zhang et al. [55]	2011	rat	human	stereotaxic injection of human NM into left SN	Microgliosis and loss of TH-positive neurons
Zecca et al. [56]	2008	rat	human	local injection of human NM into SNpc	Microglia activation, decreased number of TH-positive neurons no changes in DA levels in the striatum 1 week post injection
Double et al. [57]	2003	rat	human	injection of iron-loaded NM and/or lisuride into left SNpc	NM induces degeneration of TH-positive neurons lesions, prevented by lisuride, no changes in DA levels in striatum, 25% decrease in HVA in Fe-NM treated animals, no behavioral deficits

## Data Availability

No new data were created or analyzed in this study. Data sharing is not applicable to this article.

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
