# Peer review of "The Enigma of Neuromelanin: How Animal Models Can Help Researchers to Understand the Link Between Pigments and Neurodegeneration"

_biomedicines, 2025, doi:10.3390/biomedicines13123048_

Round 1
Reviewer 1 Report
Comments and Suggestions for Authors
This contribution presents an excellent review that examines the role of neuromelanin (NM) and neuromelanin granules (NMGs) within the context of Parkinson’s disease pathophysiology. It is a timely contribution, since NM has recently gained momentum. Although the potential link between NM and dopaminergic cell vulnerability has been postulated a long time ago, current appeal is possible due to the recent introduction of pigmented animal models (transgenic and AAV-mediated, comprising both rodents and non-human primates).
NM is a dark pigment located in the cytoplasm of dopaminergic neurons, likely resulting from the non-enzymatic auto-oxidation of dopamine (although this issue does not fully explain the lack of NM in rodents, as explained here in this review).
Please find a few minor comments, intended to further improve the narrative:
1. Pigmented models of PD based on the AAV-mediated enhanced expression of tyrosinase are here taken as ‘transgenic rodent models’ (page 6, line 244). In a strict sense, ‘transgenic animals’ involve the introduction of a gene into the pronucleus of fertilized eggs, which are then transferred into the oviducts of pseudopregnant females. The injected DNA is integrated into the mouse genome and is normally segregated in the progeny. Founder mice generated in this way are mated, giving rise to stable pedigrees expressing the transgene. Not that this does not apply to animal models generated by the viral vector-mediated overexpression of tyrosinase. Accordingly, and instead of ‘transgenic animals’, it makes more sense to consider these animals as ‘AAV-mediated’ or anything similar.
2. Reference No. 63 should be cited as follows: Chocarro J, et al., Introducing PIGMO, a novel PIGmented Mouse model of Parkinson’s disease (V1), bioRxiv 2025, https://doi.org/10.1101/2025.05.09.653000
3. iPSCs and NM (lines 265-281): studies performed in fibroblasts from parkinsonian patients carrying a homozygous DJ-1 mutation (PARK7 gene), reprogrammed into iPSCs and then differentiated into dopaminergic neurons, revealed elevated mitochondrial stress by day 50, together with NM accumulation resulting from dopamine oxidation. Compared to dopaminergic cells generated from patients carrying familial mutations, only those with DJ-1 mutations exhibited NM pigmentation. See Burbulla LF, Song P, Mazzuli JR, Zampese E, Wong YC, Jeon S, et al. (2017) Dopamine oxidation mediates mitochondrial and lysosomal dysfunction in Parkinson’s disease. Science 357:1255-1261. https://doi.org/10.1126/science.aam9080
4. Subheading 3.5 deals with the relationship between NMGs and Lewy bodies (LBs). This is a very interesting topic. Although available pigmented models postulated that NM accumulation beyond a given threshold triggers the aggregation of endogenous alpha-synuclein in the form of LBs, mechanistic insights explaining the interplay between NM and LBs still need to be identified. In this regard, both in pigmented models as well as in post-mortem samples taken from PD patients, LBs were only found in pigmented neurons remaining alive at the time of the necropsies. In other words and regarding dopamine cell vulnerability, it might be the case that LBs cannot be taken as the villains, NM being considered as the butler.
5. Line 505: ‘elucidate’ instead of ‘Elucidate’
Reviewer 2 Report
Comments and Suggestions for Authors
- Please mention the NIGRASOMES documented in some recent papers and concerning neuromelanin. It has not been mentioned in the present version of the reviewed MS.
- The paper concerns substantia nigra and the related neurions, however neuromelanin is also produced in locus coeruleus, and in some parts of labyrinh and inner ear, where its functions are discussed. Please supplement the text.
- The meninges also contain melanotic cells, which are most probably melanocytes. These cells may be a source of the so called aseptic dermatitis. This melanin is probably eumelanin, but eumelanin may be produced also in other parts of brain, and this fact may affect the conclusions of your review paper. This fact may be somehow addressed in your text. (Goldgeier et al., J Investigatative Dermatology, 82:235-238, 1984, and other papers of this subject)
- It is difficult to call the bacterial pigment neuromelanin, as bacteria do not reveal the candidate for neural system. Its functioning in bacteria is postulated, and the types of bacterial melanin are at present deeply searched. Any new discovery in this field, in particular concerning pheomelanin should me mentioned in this review.
- As citing prof Zecca et al., I cannot find the corresponding papers e.g. by , D'Ischia, Napolitano and other scientists who study melanins, including neuromelanins. In particular, some papers on eye melanin of retinal pigmented epithelium and uvea, is studied and the eye is actually the part of brain (particularly the inner pigmented epithelium - and the retina), these facts may be found in this review paper to get the full view of the problem.
- The authors mentioned some biophysical methods including MRI and NMR, but the application of electron paramagnetic (spin) resonence (EPR, ESR) has been fully omitted. This technique is by choice one of the most important faminy of biophysical techniques and in the opinion of the present reviewer, it should be mentioned, in particular as a method to study the described animal models.
